# Reversible induction of mitophagy by an optogenetic bimodular system

Pasquale D'Acunzo [1], Flavie Strappazzon [2,3], Ignazio Caruana [1], Giacomo Meneghetti [4], Anthea Di Rita [2,3], Luca Simula [1,3], Gerrit Weber [1], Francesca Del Bufalo [1], Luisa Dalla Valle [4], Silvia Campello [2,3], Franco Locatelli [1,5] & Francesco Cecconi [1,3,6]

Autophagy-mediated degradation of mitochondria (mitophagy) is a key process in cellular quality control. Although mitophagy impairment is involved in several patho-physiological conditions, valuable methods to induce mitophagy with low toxicity in vivo are still lacking. Herein, we describe a new optogenetic tool to stimulate mitophagy, based on light-dependent recruitment of pro-autophagy protein AMBRA1 to mitochondrial surface. Upon illumination, AMBRA1-RFP-sspB is efficiently relocated from the cytosol to mitochondria, where it reversibly mediates mito-aggresome formation and reduction of mitochondrial mass. Finally, as a proof of concept of the biomedical relevance of this method, we induced mitophagy in an in vitro model of neurotoxicity, fully preventing cell death, as well as in human T lymphocytes and in zebrafish in vivo. Given the unique features of this tool, we think it may turn out to be very useful for a wide range of both therapeutic and research applications.

[1] Department of Paediatric Haematology, Oncology and Cell and Gene Therapy, IRCCS Bambino Gesù Children's Hospital, Piazza Sant'Onofrio 4, 00165 Rome, Italy. [2] IRCCS Fondazione Santa Lucia, Via del Fosso di Fiorano 64, 00143 Rome, Italy. [3] Department of Biology, University of Tor Vergata, Via della Ricerca Scientifica 1, 00133 Rome, Italy. [4] Department of Biology, University of Padova, Via Ugo Bassi 58/b, 35131 Padova, Italy. [5] Department of Gynecology/ Obstetrics and Pediatrics, Sapienza University of Rome, Piazzale Aldo Moro 5, 00185 Rome, Italy. [6] Unit of Cell Stress and Survival, Danish Cancer Society Research Center, Strandboulevarden 49, DK-2100 Copenhagen, Denmark. Correspondence and requests for materials should be addressed to F.C. (email: cecconi@cancer.dk)

Autophagy-mediated degradation of mitochondria (hereafter mitophagy) is a pivotal quality control mechanism in cellular homeostasis[1]. Briefly, in normal conditions, aged and damaged mitochondria are ubiquitylated and engulfed in double membrane vesicles called autophagosomes (APs), which, in turn, are transported and fused to lysosomes in order to release their cargo. Given the importance of mitochondria in adenosine triphosphate (ATP) production, calcium buffering, redox reactions, reactive oxygen species (ROS) generation, and death/survival choice[2], cells need to finely regulate the turnover of these organelles to maintain internal stability. Accordingly, mitophagy defects have been implicated in the initial steps of several diseases, such as neurodegenerative diseases, muscle atrophy, and carcinogenesis, in which this housekeeping process is strongly downregulated[3].

Nonetheless, valuable methods to selectively and reversibly induce mitophagy with low-level side effects are still lacking, restraining the study of mitophagy to selected cases and conditions.

In conventional cell biology studies, the most-widely used strategy encompasses the dissipation of the $H^+$ proton gradient across the inner mitochondrial membrane, through administration of uncoupling agents—carbonyl cyanide-4-(trifluoromethoxy)phenylhydrazone (FCCP), 2,4-dinitrophenol (2,4-DNP or simply DNP), etc.—or electron transport chain inhibitors (oligomycin/antimycin-A). Accordingly, uncouplers cause rapid depolarization of mitochondrial potential ($\Delta\Psi$m) and mitochondrial damage. Consequently, E3 ubiquitin ligases, such as Parkin, are recruited to depolarized mitochondria, where they ubiquitylate their substrates and induce mitochondrial clearance[2].

Administration of these compounds carries several disadvantages. First of all, they show a broad spectrum of off-target activities, e.g., plasma membrane depolarization[4], ATP production block[5], mitochondrial permeability transition pore opening[6], cytotoxicity[7] and, ultimately, cell death[8–10]. Second, uncoupler treatments are not suitable in vivo, since the fast $H^+$ influx into the mitochondrial matrix is responsible for strong hyperthermia in mammals[11]. Third, mitophagy activation by $\Delta\Psi$m depolarization seems to require PINK1/Parkin activity, at least in a number of model systems[12]. This pathway, however, has been found to be mutated or impaired in some diseases, such as Parkinson's disease (PD)[13].

One way, usually followed, to overcome some of these issues had been the genetic manipulation of specific genes along the mitophagy pathway. Downregulation of the mitochondrial deubiquitinase USP30, for instance, has been shown to provoke a strong mitophagy response with low toxicity, and was able to counteract oxidative stress-driven neurotoxicity in vivo in *Drosophila melanogaster*[14]. Likewise, the forced relocalization of the pro-autophagy protein AMBRA1 (autophagy and beclin-1 regulator 1) to the mitochondrial outer membrane (MOM) directly stimulates mitophagy[15]. In fact, endogenous AMBRA1 can be found at mitochondria[16] and is able to directly recruit APs to damaged organelles through Microtubule-associated proteins 1A/1B light chain 3B (LC3) binding upon stress induction[15]. When fused C-terminally to a small signal peptide from the *Lysteria monocytogenes* ActA (actin assembly inducing) protein, it could be relocalized to the MOM[15], where it induces mitophagy per se, in the absence of any other stimulus, in both Parkin-dependent or -independent ways[15]. Notably, AMBRA1-ActA-mediated mitophagy was sufficient to alleviate oxidative stress and significantly reduce cell death in commonly used in vitro models of PD, namely in rotenone and 6-hydroxydopamine(6-OHDA)-intoxicated neuroblastoma cells[17].

Although genetic manipulation led to good results in terms of toxicity and specificity, in practice it is rarely used as mitophagic tool, since the cellular response is hardly tuneable and cannot be switched off.

Herein, we present an optogenetic bimodular system, based on the recruitment of AMBRA1 to mitochondria after blue light irradiation, which stimulates mitophagy in a specific and reversible fashion. As a proof of concept, we demonstrate effective mitophagy induction (I) in vitro, in HeLa cells, which are worldwide considered a Parkin-free cell line[18], (II) ex vivo, in human T lymphocytes collected from peripheral blood of healthy donors, and (III) in vivo, in illuminated living *Danio rerio* embryos. Moreover, we also show a light-dependent block of apoptosis in an in vitro model of oxidative stress-mediated proneural-like cell death.

Besides its relevance as a putative therapeutic tool, this is a formidable example of the potential application of optogenetic dimers to mediate not easily tuneable cellular processes in an efficient and reversible way.

## Results

**AMBRA1 is recruited to the MOM upon blue light stimulation.** To date, diverse blue light-induced dimerizers have been described; among others, we have chosen the iLID/sspB$_{micro}$ system[19]. In fact, this system is characterized by fast kinetics, large changes in binding affinity upon irradiation, low (if any) binding in the dark state, no requirement of exogenous chromophores and very low-molecular weight of the modules, so making it one of the most attractive optogenetic pairs hitherto published[20].

Thus, we N-terminally fused the sequence encoding AMBRA1 to the already published TagRFPt-sspB$_{micro}$ protein (hereafter, AMBRA1-RFP-sspB). Its counterpart (Venus-iLID-ActA) carried a more stable modified version of the green fluorescent protein (Venus), the blue-light sensor element iLID and the MOM signal peptide of the ActA protein (Fig. 1a). In our working model, AMBRA1-RFP-sspB is present in the cytosol in a basal state, while its partner is tethered to the mitochondrial membrane. Upon blue light stimulus, iLID undergoes a conformational change which undocks the ssrA peptide and allows its binding to the sspB stretch[19] (Fig. 1a). The expected final result would be a blue-light inducible shuttling of AMBRA1 from its main diffuse cytosolic localization to the MOM; this event should then trigger mitophagy without additional stimuli, de facto reproducing the phenotype we obtained by overexpressing AMBRA1-ActA in different cell types[15]. Note that the local irradiance necessary for sspB/iLID binding (500 µW per $cm^2$)[20] is lower, by several orders of magnitude, than that used for classical optogenetic opsins (~1 mW per $mm^2$, corresponding to $10^5$ µW per $cm^2$)[21], strongly limiting unspecific effects and photodamage in prolonged experiments.

To confirm our hypothesis, Venus-iLID-ActA was transfected alone in HeLa cells, which were, in turn, kept in the dark or illuminated for 72 h at an irradiance of 500 µW per $cm^2$ with a blue pulsed light LED emitter (cycles of 1 s of light spaced by 1 min of dark resting state). Cells showed a healthy, fused and elongated mitochondrial network, even after prolonged pulsed illumination, this suggesting that neither overexpression of the MOM-tagged Venus-iLID-ActA nor blue light exposure were altering per se mitochondria dynamics. Moreover, as expected, Venus-iLID-ActA was found to entirely co-localize with the MOM-marker Tom20 (Supplementary Figure 1).

Next, Venus-iLID-ActA and AMBRA1-RFP-sspB were coexpressed and complete recruitment of AMBRA1-RFP-sspB to mitochondria in irradiated cells was obtained (Fig. 1b–d). First, we analyzed the subcellular distribution of AMBRA1-RFP-sspB by Western blot (WB) assay, by separating the mitochondrial

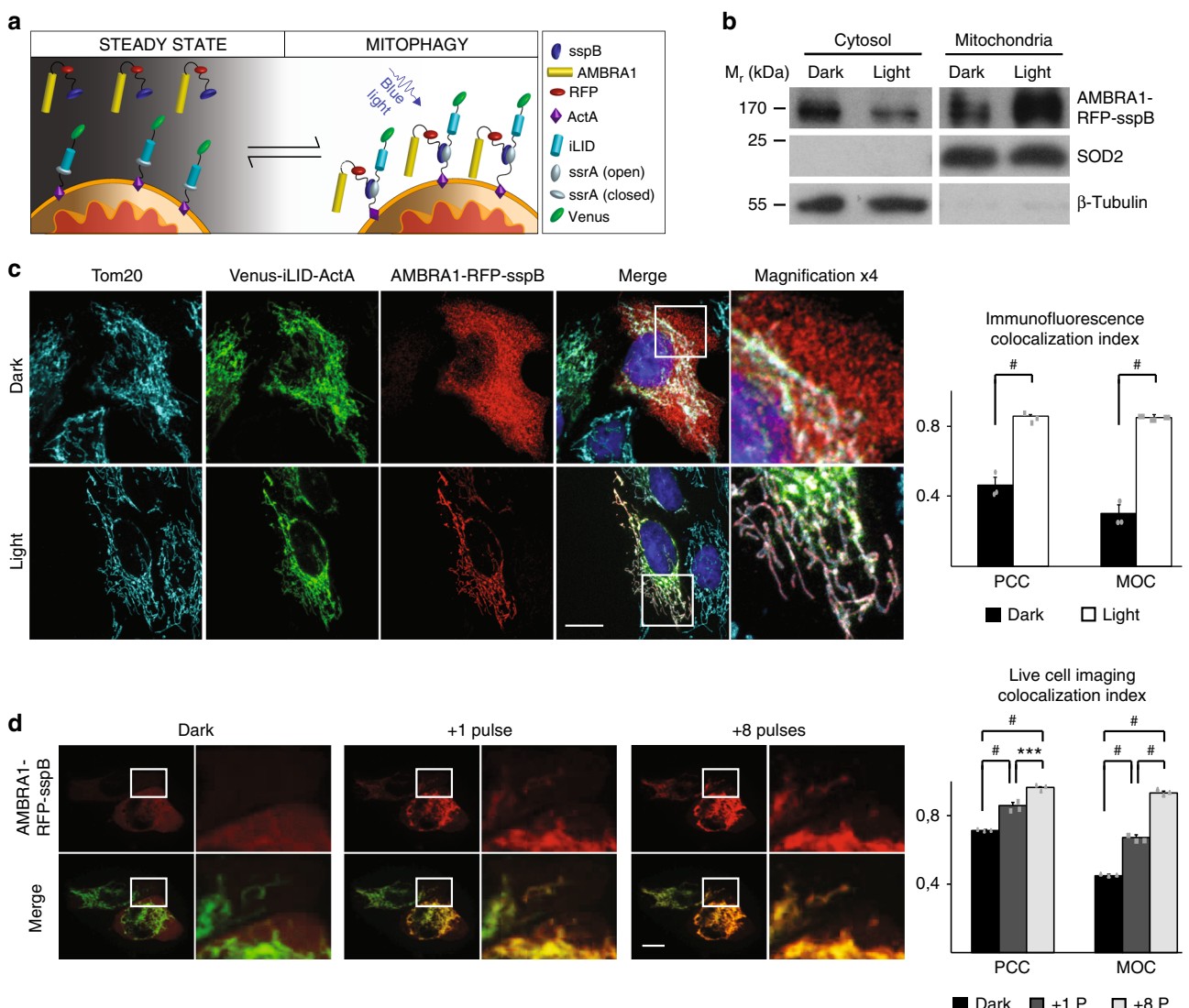

**Fig. 1** AMBRA1-RFP-sspB is relocalized to MOM upon blue light exposure. **a** Scheme of the blue light-dependent, AMBRA1-RFP-sspB-mediated induction of mitophagy. In resting conditions, Venus-iLID-ActA is tethered to the MOM, while AMBRA1-RFP-sspB is found in the cytosol (STEADY STATE panel, left). Upon blue light administration, iLID undergoes into a conformational change, which unmasks the ssrA peptide and permits the high-affinity binding between ssrA and sspB (MITOPHAGY panel, Right). Thus, the pro-autophagy protein AMBRA1 (covalently bound to sspB) accumulates to the edge of the MOM where it promotes autophagy-mediated clearance of mitochondria (mitophagy). **b** HeLa cells, transfected with plasmids encoding Venus-iLID-ActA/ AMBRA1-RFP-sspB, were exposed to continuous blue light for 30 s or kept in the dark. Crude mitochondrial and cytosolic extracts were analyzed by WB through an anti-AMBRA1 antibody to reveal AMBRA1-RFP-sspB. SOD2 and β-tubulin were used as loading control for mitochondrial and cytosolic lysates, respectively. $M_r$ (kDa): relative molecular mass expressed in kilodalton. **c** HeLa cells were transfected and treated as described in (**b**). Subsequently, cells were fixed and immuno-stained with antibodies against AMBRA1 (red) and Tom20 (MOM marker, cyan). The green signal shown in the figure is the intrinsic fluorescence of the Venus-iLID-ActA protein. Pearson's correlation coefficient (PCC) and Manders' overlap coefficient (MOC) of the red over the green signal were quantified in ten random fields of three independent experiments. Nuclei (blue) were stained with DAPI. **d** HeLa cells overexpressing Venus-iLID-ActA/AMBRA1-RFP-sspB were filmed through the UltraVox (PerkinElmer) live cell imaging spinning disk microscope before (Dark in the panel) and during 8 irradiation cycles, consisting of 1 pulse (50 ms) of blue light followed by 35 s of dark resting state. The graph shows PCC and MOC quantifications of the red over the green signal for three conditions (Dark, one pulse, eight pulses) in ten random fields of three independent experiments. Images are the sum of a three frames Z-stack. Insets: 4× magnification. Scale bars: 10 μm. Data shown: mean ± S.E.M. Hypothesis tests: Student's $t$ test in (**c**) and ANOVA test in (**d**). $^{***}p < 10^{-3}$. $^{\#}p < 10^{-4}$. Source data are provided as a Source Data file

from the cytosolic protein pools in Venus-iLID-ActA/AMBRA1-RFP-sspB-overexpressing cells lysates. Upon blue light exposure, the AMBRA1-RFP-sspB signal clearly decreased into the cytosolic fractions whilst increased in the mitochondrial one (Fig. 1b). Second, AMBRA1-RFP-sspB relocalization after a single burst of blue light was confirmed in fixed cells incubated with antibodies against Tom20 and AMBRA1. In cells kept in the dark, AMBRA1-RFP-sspB was found in the cytosol, while it was almost completely co-localized with Tom20 and Venus-iLID-ActA after stimulus—Pearson's correlation coefficient (PCC): 0.86 ± 0.04 (mean ± standard deviation); Manders' overlap coefficient (MOC): 0.85 ± 0.05 Fig. 1c. We also report 9.4 ± 1.4% of cells in which a faint signal was retained in the cytoplasm upon illumination, mostly in Venus-iLID-ActA low-expressing cells (Supplementary Figure 2). Last, to better understand the binding kinetics in our conditions, we recorded AMBRA1-RFP-sspB

shuttling from cytosol to mitochondria within the same cell, by live cell imaging (Supplementary Movie 1). A single 50 ms pulse was enough to efficiently drive the vast majority of AMBRA1-RFP-sspB to mitochondria (MOC: $0.67 \pm 0.04$ vs. $0.45 \pm 0.02$, ANOVA test, $p < 10^{-4}$); nevertheless, a rising number of spikes permitted a slightly increase of the co-localization rate. After 8 spikes, the translocation of the protein reached the plateau (MOC: $0.93 \pm 0.04$, ANOVA test, $p < 10^{-4}$ Fig. 1d).

**Prolonged AMBRA1 repositioning to the MOM induces mitophagy.** Next, we wondered whether the persistent presence of AMBRA1-RFP-sspB on the MOM was able to trigger mitophagy as already observed in the case of AMBRA1-ActA ectopic expression[15]. HeLa cells (a bona fide Parkin-free cell line) were transfected with the two plasmids encoding Venus-iLID-ActA and AMBRA1-RFP-sspB and irradiated with pulsed blue light for 30 min, 1 h, 2 h, or 4 h to monitor the status of their mitochondria. In the given conditions, sporadically after 30 min and more prominently after 4 h, mitochondria appeared fragmented, round-shaped and accumulated next to the nuclei, usually in 1 or 2 opposite poles (Fig. 2a), with these structures strongly resembling mito-aggresomes, a hallmark of ongoing mitophagy[22]. The appearance of AMBRA1-RFP-sspB rings around mito-aggresomes was also evident at higher magnification (Fig. 2a, inset), as already reported for AMBRA1-ActA[15]. Of note, as an internal control, nontransfected neighbor cells showed normal-looking mitochondria, suggesting that the effect described is not due to unspecific phenomena related to illumination. Upon light exposure the percentage of transfected cells with mito-aggresomes increased over time, raising from $10.8 \pm 1.5\%$ to $31.9 \pm 4.5\%$, while the extent of normal cells decreased accordingly (Supplementary Figure 3). When pulsed irradiation was prolonged to 24 h, more evident phenotypes were detected, as $13.4 \pm 3.7\%$ of double-transfected cells had only mitochondrial remnants (Supplementary Figure 4). Remarkably, little (if any) pyknosis is seen by DAPI staining of irradiated Venus-iLID-ActA/AMBRA1-RFP-sspB cells (Fig. 2a, Supplementary Figure 3, Supplementary Figure 4), suggesting lack of apoptosis when mito-aggresomes are formed. To better elucidate this point, we analyzed the percentage of Venus-iLID-ActA/AMBRA1-RFP-sspB double-positive viable cells by flow cytometry when either kept in the dark or illuminated for 24 h, finding no differences (Supplementary Figure 5); moreover, at a single-cell level, no release of cytochrome-c from mito-aggresomes was documented (Supplementary Figure 6). These data suggest that AMBRA1-RFP-sspB-mediated mitophagy was not followed by cell death in our conditions.

Subsequently, to better understand if upon illumination the autophagy machinery led to a reduction in mitochondrial mass, we assessed by WB the amount of three distinct mitochondrial markers in cells treated with the lysosome inhibitor ammonium chloride ($NH_4Cl$, 40 mM) and in the presence or absence of pulsed blue light for 24 h (Fig. 2b, left panel). More specifically, we analyzed the signal of Tom20 (MOM protein), and two mitochondrial matrix proteins, SOD2 and HSP60. As previously described[23], mitochondrial matrix proteins are the best markers to estimate mitochondrial mass since they are preferentially degraded by mitophagy. Indeed, all markers were reduced in irradiated cells compared to their dark counterparts, while no changes were documented in $NH_4Cl$-treated cells, strongly suggesting that reduction of the signal mediated by AMBRA1-RFP-sspB translocation to the MOM is mitophagy dependent. As a negative control, we repeated the same experiment in Venus-iLID-ActA/RFP-sspB co-expressing cells. RFP-sspB retains the ability to bind its partner upon blue light exposure, but it does not carry the pro-autophagy effector protein AMBRA1. As expected,

no alteration in mitochondrial morphology and no decrease in mitochondrial markers were detected after illumination (Fig. 2b, right panel, Supplementary Figure 7 and Supplementary Figure 8). Since wild-type HeLa cells are Parkin-deficient and almost insensitive to mitophagy-inducing insults without overexpression of Parkin or other mitophagic receptors[24], a direct comparison between our optogenetic tool and pre-existing methods has been prohibitive for this cell line. However, in Parkin-competent cells (HEK293T) the AMBRA1-RFP-sspB-mediated, light-induced reduction in mitochondrial mass was comparable to that triggered with classical methods, as the Oligomycin/Antimycin A treatment (Supplementary Figure 9). Next, In order to corroborate the relationship existing between the reduction of mitochondrial markers and autophagy induction, we performed a pulse-chase experiment in which a modified methionine analogue (L-azidohomoalaine, AHA) has been incorporated into nascent proteins before the illumination procedures. After 24 h of pulsed blue light illumination or 24 h of dark resting state, AHA levels have been quantified and a significant decrease was reported in Venus-iLID-ActA/AMBRA1-RFP-sspB double-transfected and irradiated cells when compared to control cells that were treated in the same way (Supplementary Figure 10). Given that a long chase time (24 h in our experiments) guaranteed the exclusive measurement of the long-lived proteins proteolytic rate, our data strongly suggest that blue light irradiation stimulated autophagy specifically in Venus-iLID-ActA/AMBRA1-RFP-sspB co-expressing cells.

Finally, mitochondrial rearrangement and reduction during AMBRA1-mediated mitophagy was recorded over time by live cell imaging (Supplementary Movie 2, Supplementary Figure 11 and Fig. 2c). Since Venus-iLID-ActA is constantly anchored to the MOM, live visualization over time of this protein in not-bleaching conditions makes it possible both to estimate mitochondrial mass within single cells (proportional to Venus-iLID-ActA fluorescence) and to monitor how mitochondria behave when AMBRA1 shuttles to their surface. A progressive remodeling of mitochondria was appreciable, this leading to a fragmented perinuclear signal. Mitochondrial aggregation was also confirmed by the reduction of the area occupied by mitochondria (from $123 \pm 2$ to $25 \pm 6 \, \mu m$, ANOVA test, $p < 10^{-3}$). While the mean signal did not decrease (ensuring no photobleaching), the decrease of green fluorescence per whole cell was also evident (from a coefficient of $8.9 \pm 0.1 \times 10^5$ down to $2.3 \pm 0.6 \times 10^5$, ANOVA test, $p < 10^{-3}$) again suggesting ongoing mitochondrial clearance in our given conditions.

**AMBRA1-RFP-sspB-mediated mitophagy is reversible.** Venus-iLID/RFP-sspB binding is reversible, as previously described[19] and as confirmed by us (Supplementary Figure 12). Therefore, by live cell imaging, we set up to establish whether our fusion proteins maintained this important property and conferred the same feature to the phenotype. To this aim, we first stimulated AMBRA1-RFP-sspB repositioning to MOM in Venus-iLID-ActA/AMBRA1-RFP-sspB overexpressing cells; then, we examined AMBRA1-RFP-sspB shuttling when the dark state was restored. After 3 min, the AMBRA1-related mitochondrial signal was highly weakened, while the cytosolic one had an opposite tendency, revealing an ongoing detachment of AMBRA1-RFP-sspB from mitochondria (Fig. 3a, Supplementary Figure 13, Supplementary Movie 3). However, a new burst of blue light was able to fully bring back the protein to the former position, thus proving that AMBRA1-RFP-sspB displacement could be precisely regulated.

We next checked if a continuous dark state could be sufficient to thwart previous induction of mitophagy by pulsed light, in

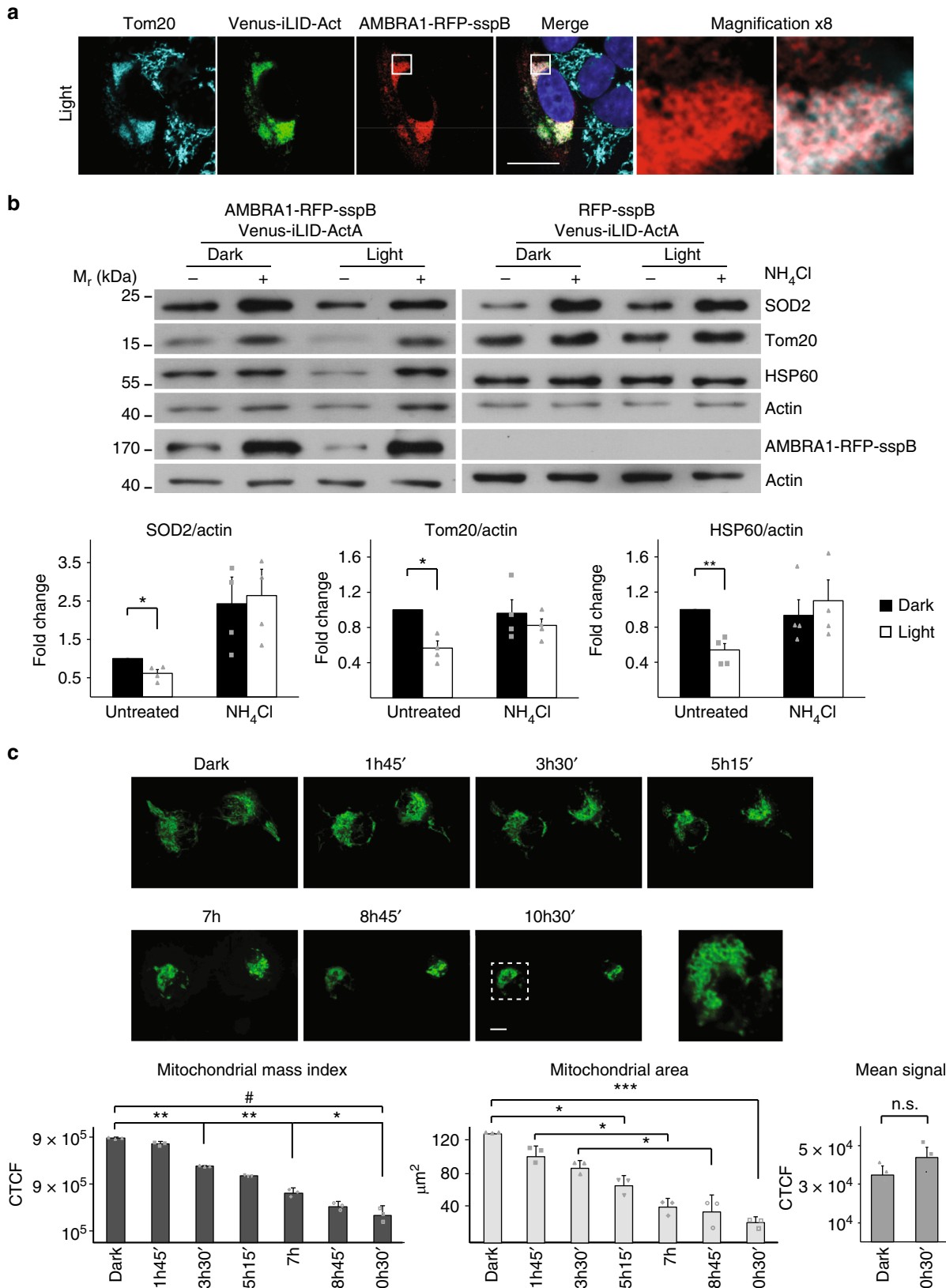

Venus-iLID-ActA/AMBRA1-RFP-sspB co-expressing cells. Hence, cells were illuminated for 24 h followed by 24 h of dark (a condition that we termed Rescue). Then, we compared the levels of the mitochondrial marker SOD2 with those derived from cells treated in the opposite way (24 h resting, then 24 h light) in order to estimate mitochondrial mass in the two given conditions. As controls, cells were also kept either 48 h in the dark or 48 h under blue light, respectively; additionally, the experiment was repeated in Venus-iLID-ActA/RFP-sspB co-expressing cells to highlight the specificity of SOD2 decrease, as explained before

**Fig. 2** AMBRA1-RFP-sspB shuttling to the MOM induces mitophagy. **a** Venus-iLID-ActA/AMBRA1-RFP-sspB overexpressing HeLa cells were irradiated or not for 4 h with pulsed, blue light. Cells were subsequently fixed and stained for Nuclei (DAPI, blue), AMBRA1 (red) and Tom20 (cyan). Inset: 8× magnification. Scale bar: 10 μm. **b** HeLa cells, transfected with plasmids encoding Venus-iLID-ActA/AMBRA1-RFP-sspB (left panel), were stimulated with a blue LED irradiator or left in the dark for 24 h. In the meantime, they were treated or not with 40 mM NH₄Cl, a lysosome inhibitor. As a negative control, the same experiment was repeated in Venus-ILID-ActA/RFP-sspB co-expressing cells (right panel). In the subsequent WB analysis, Tom20, SOD2, and HSP60 were used as mitochondrial markers, while actin was the loading control. AMBRA1-RFP-sspB was detected to verify the rate of overexpression. Graphs recapitulate the normalized ratio between the densitometric signals of the three mitochondrial markers over actin in four independent experiments involving Venus-iLID-ActA/AMBRA1-RFP-sspB overexpressing cells. For the quantification of the same parameters in the negative control see Supplementary Figure 7. Data shown: mean ± S.E.M. Hypothesis test: Student's $t$ test. $*p < 5 \times 10^{-2}$. $**p < 10^{-2}$. $M_r$ (kDa): Relative molecular mass expressed in kilodalton. **c** Single-HeLa cells co-expressing Venus-iLID-ActA/AMBRA1-RFP-sspB were followed in time for 10 h 30 min during mitophagy progression through live cell imaging of the protein Venus-iLID-ActA (50 ms blue laser spikes alternated by 1 min dark). Each frame depicts mitochondria morphology every 1 h 45 min of stimulation. Images are the sum of a three frames Z-stack. Graphs show the area occupied by mitochondria over time, the overall reduction per whole cell of the Venus-iLID-ActA signal intensity corrected for the background and the mean fluorescence intensity in three independent experiments. Inset: 4× magnification of a single plane after 10 h 30 min of stimulation, highlighting mitoaggresome formation. CTCF corrected total cell fluorescence. Scale bar: 10 μm. Data shown: mean ± S.E.M. Hypothesis test: ANOVA test for Area and Mito Mass Index, Student's $t$ test for the mean signal. n.s. not statistically significant. $*p < 5 \times 10^{-2}$. $**p < 10^{-2}$. $***p < 10^{-3}$. $\#p < 10^{-4}$. Source data are provided as a Source Data file

(Fig. 3b). Strikingly, the Rescue condition permitted the re-establishment of normal SOD2 levels, which were significantly higher than what observed in the 24 h lane, this strongly suggesting a complete halt in mitophagy progression. As expected, no effect was observed for Venus-iLID-ActA/RFP-sspB co-expressing cells.

Last, in order to further elucidate the molecular mechanisms at the basis of AMBRA1-RFP-sspB-mediated mitophagy in a Parkin-independent context, we checked whether inhibition of the HUWE1 E3 Ubiquitin Ligase or the IKKα kinase (two key factors in AMBRA1-dependent mitophagy[24]) were able to block mitophagy in HeLa cells. As expected, both the downregulation of HUWE1 and the pharmacological irreversible inhibition of IKKα were able to fully prevent reduction of mitochondrial mass upon blue light stimulus in Venus-iLID-ActA/AMBRA1-RFP-sspB co-expressing HeLa cells (Supplementary Figure 14). This suggests that the HUWE1/IKKα-axis is a crucial pathway to fully trigger AMBRA1-RFP-sspB-mediated mitophagy.

**Optogenetic mitophagy induction in physiological contexts.** In order to assess if our bimodular system could be used in other applications besides in vitro studies, we checked whether mitophagy could be induced in ex vivo human T lymphocytes taken from healthy donors and in living animals in vivo. To this aim, we co-infected T cells with retroviral vectors encoding for Venus-iLID-ActA/AMBRA1-RFP-sspB or Venus-iLID-ActA/RFP-sspB as a control (Supplementary Figure 15) and illuminated or not these cells for 24 h by pulsed blue light as indicated above. AMBRA1-RFP-sspB shuttling to mitochondria as well as a reduction in mitochondrial mass was confirmed also in this experimental condition (Fig. 4). AMBRA1-RFP-sspB but not RFP-sspB-expressing cells showed a marked decrease in Tom20 staining when illuminated (from a coefficient of $31.7 \pm 3.1$ to $7.6 \pm 1.6$, $p < 10^{-4}$, Student's $t$ test, Fig. 4a) as well as decreased mitochondrial markers by WB (Fig. 4b), with this indicating blue light-driven mitochondrial clearance.

For in vivo stimulation of AMBRA1-RFP-sspB-mediated mitophagy, we chose zebrafish (*D. rerio*) as a model organism for several reasons. Technically, the optical transparency of zebrafish embryos permitted an easier penetration of blue light when compared to rodents. Functionally, it has already been demonstrated that AMBRA1 is highly conserved in this species[25], and that injection of human *AMBRA1* mRNA is able to rescue the phenotype of zebrafish morphants in which endogenous genes were knocked-down[26], strongly suggesting overlapping functions across species. Intriguingly, zebrafish *ambra1a* knockdown larvae display accumulation of swollen and aberrant mitochondria in

skeletal muscle fibers[26], while the protein sequence is predicted in silico[27] to carry a LC3-interacting region domain in its C-terminal region (Supplementary Figure 16), similarly to human AMBRA1[15]; this clearly indicates a possible role for ambra1a in zebrafish mitophagic pathways.

In view of that, we microinjected Venus-iLID-ActA/AMBRA1-RFP-sspB expressing plasmids in zebrafish embryos and checked whether mitophagy could be induced by blue light. Preliminarily, we demonstrated a proper MOM distribution for Venus-iLID-ActA alone (Fig. 5a) and a correct mitochondrial shuttling of AMBRA1-RFP-sspB from and to mitochondria of zebrafish muscle fibers with similar kinetics observed in human cells in vitro (MOC of $0.877 \pm 0.09$ after light, down to $0.402 \pm 0.11$ when dark was restored, $p < 10^{-2}$, ANOVA test, Fig. 5b). Lastly, double-positive Venus-iLID-ActA/AMBRA1-RFP-sspB or Venus-iLID-ActA/RFP-sspB (negative control) were analyzed by live cell imaging before and after 8 h of pulsed blue light stimulation (Fig. 5c). Upon light stimulus, mitochondria appeared round-shaped (arrowheads), while Venus-iLID-ActA intensity per single fiber was significantly decreased, suggesting ongoing mitochondrial clearance. No changes could be detected in Venus-iLID-ActA/RFP-sspB-positive embryos.

**Mitophagy prevents cell death in a model of neurotoxicity.** As a proof of concept of the potential beneficial contribution of our bimodular system to biomedicine in a functional study, we tested whether AMBRA1-RFP-sspB-mediated mitophagy could be able to block apoptosis in a cellular model of oxidative stress-induced neurotoxicity. The role of mitochondria dysfunction as a priming event for ROS accumulation has been widely discussed in the literature, and mitophagy impairment has been postulated to be a key feature during ROS-derived neurodegeneration[28]. Indeed, a class of compounds called Parkinsonian toxins (among the others, Paraquat, rotenone, and 6-OHDA) directly impair mitochondrial activity through diverse mechanisms while being responsible of a PD-like phenotype both in vivo and in vitro[29]. Above all, the FDA-approved herbicide Paraquat (PQ; N,N′-dimethyl-4,4′-bipyridinium dichloride) has been epidemiologically linked to sporadic PD cases in humans[30], and it seems to provoke similar effects in rats[31], *Caenorhabditis elegans*[32] and *D. melanogaster*[14], as well as cell death in neuronal cell lines[33]. We already demonstrated that AMBRA1-ActA-mediated mitophagy is able per se to counteract apoptosis in parkinsonian toxins SHSY-5Y intoxicated cells;[17] moreover, the protective role of mitophagy upon PQ treatment has also been thoroughly proved in vivo, at least in *D. melanogaster*[14].

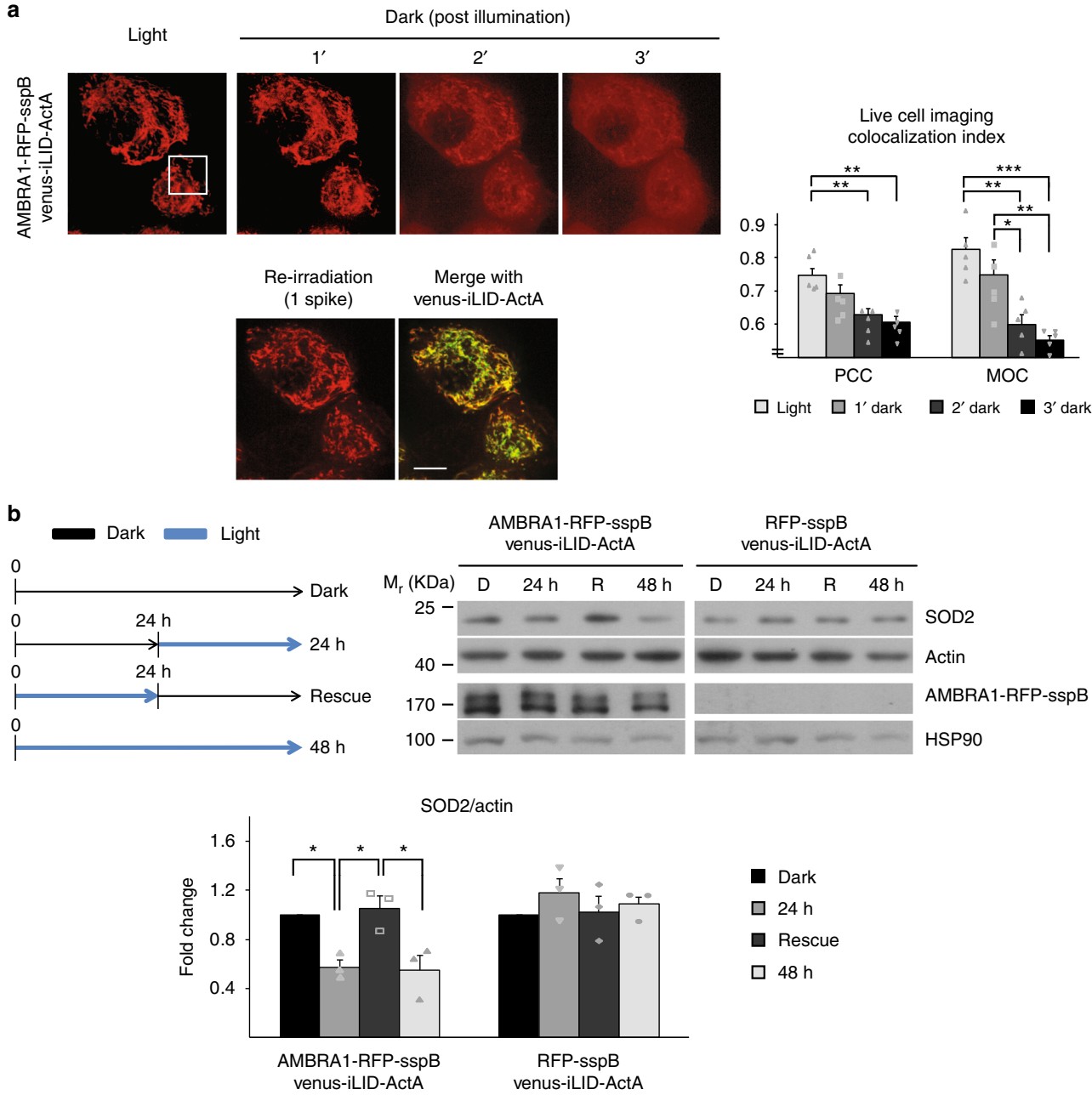

**Fig. 3** AMBRA1-RFP-sspB-mediated mitophagy is reversible. **a** AMBRA1-RFP-sspB shuttling from mitochondria was assessed by live cell imaging in single Venus-iLID-ActA/AMBRA1-RFP-sspB overexpressing HeLa cells. Upon blue light irradiation AMBRA1-RFP-sspB was found at mitochondria. Subsequently, AMBRA1-RFP-sspB subcellular distribution was recorded every minute without blue light administration. After 3 min, one spike of blue light was reapplied. Images are the sum of a three frames Z-stack. PCC and MOC of the red over the green signal were quantified in ten random fields of five independent experiments. For a ×4 magnification of the inset see Supplementary Figure 13. Scale bar: 10 μm. Data shown mean ± S.E.M. Hypothesis test: ANOVA test. $^{*}p < 5 \times 10^{-2}$. $^{**}p < 10^{-2}$. $^{***}p < 10^{-3}$. **b** Transfected HeLa cells were treated as follows: 48 h of dark (lane Dark or D), 24 h of dark + 24 h pulsed blue light (1 s light + 1 min dark, lane 24 h), 24 h of pulsed blue light + 24 h of dark state (lane Rescue or R), and 48 h of pulsed blue light only. Cell lysates were loaded on a polyacrylamide gel and immuno-blotted. Levels of the mitochondrial marker SOD2 were investigated; AMBRA1-RFP-sspB was detected to verify the rate of overexpression. Actin and HSP90 were used as loading controls of the two gels, respectively. The graph shows the normalized densitometric SOD2 over actin ratio in three independent experiments. The experiments were repeated in Venus-iLID-ActA/RFP-sspB overexpressing HeLa cells as a control (right panel). Data shown: mean ± S.E.M. Hypothesis test: ANOVA test. $^{*}p < 5 \times 10^{-2}$. $M_{r}$ (kDa): Relative molecular mass expressed in kilodalton. Source data are provided as a Source Data file

Primed by this evidence, we first verified mitophagy stimulation upon pulsed blue light exposure in Venus-iLID-ActA/ AMBRA1-RFP-sspB co-expressing embryonic telencephalic nAïve (ETNA) cells, a line derived from murine E14 striatum primordia neurons[34]. Subsequently, we investigated its protective action upon PQ-mediated damage (Fig. 6).

In these cells, mito-aggresome formation was confirmed (Fig. 6a), as well as the decrease of mitochondrial mass as measured by the protein levels of the marker HSP60 (Fig. 6b, left panel). In our system, mitochondrial damage primed by PQ seemed not to trigger mitophagy in basal conditions, whilst it robustly potentiated mitochondrial loss when

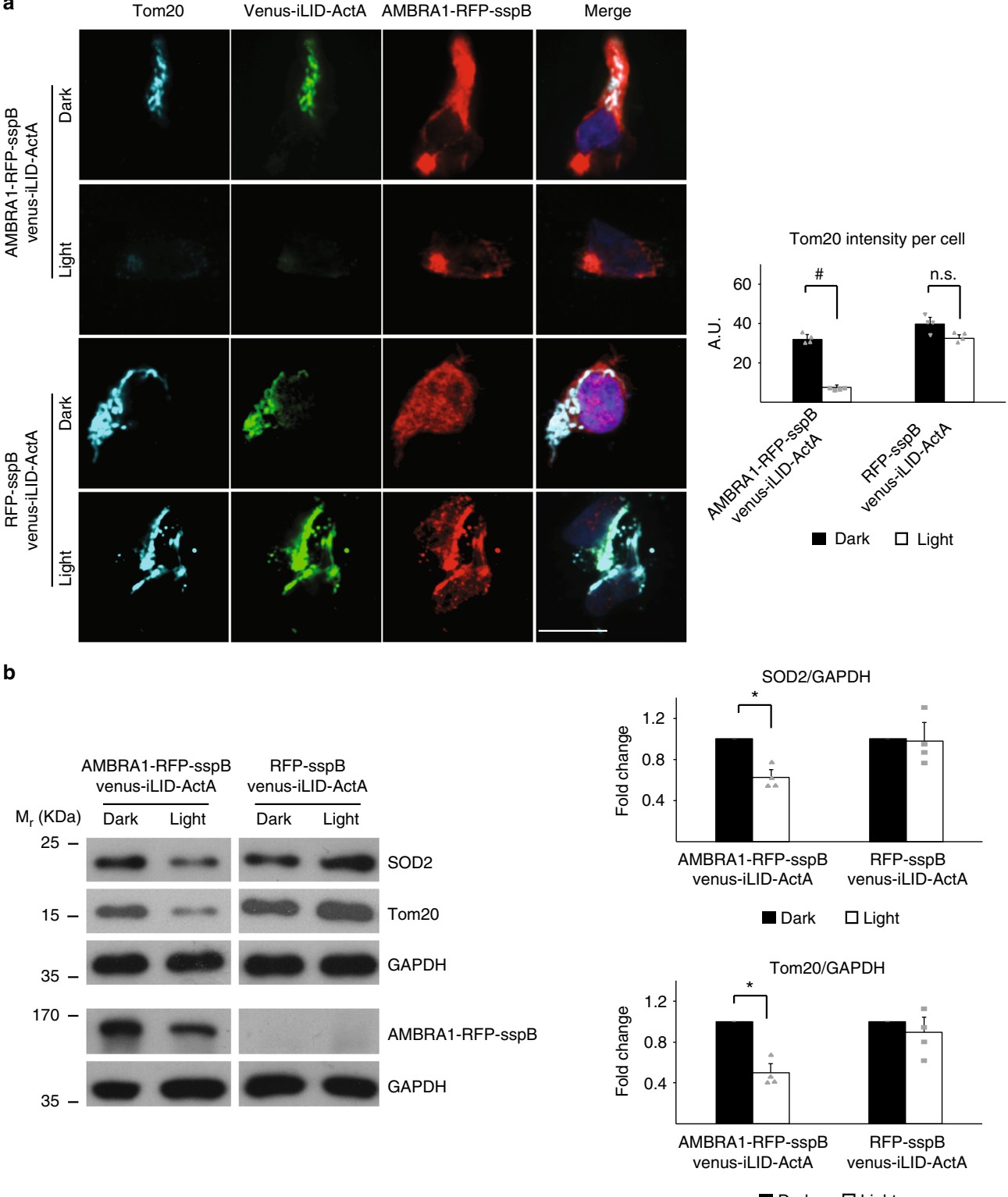

**Fig. 4** AMBRA1-RFP-sspB-mediated mitophagy can be induced in T lymphocytes. **a** Human T lymphocytes from healthy donors were double infected with viral vectors encoding Venus-iLID-ActA/AMBRA1-RFP-sspB or Venus-iLID-ActA/RFP-sspB (negative control). Cells were subsequently illuminated 24 h with pulsed (1 s light, 1 min dark) blue light or kept in the dark, then fixed and immunostained for Tom20 (cyan). Nuclei were counterstained with DAPI (blue). The graph show the intensity per cell of the Tom20 signal from four different donors. A minimum of 50 cells were analyzed per donor. Scale bar: 10 μm. Data shown: mean ± S.E.M. Hypothesis test: ANOVA test. $^{\#}p < 10^{-4}$. n.s. not statistically significant. **b** Human T lymphocytes, manipulated as described in (**a**), were lysed and analyzed by WB. Tom20 and SOD2 were used as mitochondrial markers, while GAPDH was the loading control. AMBRA1-RFP-sspB was detected to verify the rate of overexpression. Graphs recapitulate the normalized ratio between the densitometric signals of the two mitochondrial markers over GAPDH in four different donors. Data shown: Mean ± S.E.M. Hypothesis test: ANOVA test. $^{*}p < 5 \times 10^{-2}$. $M_{r}$ (kDa): Relative molecular mass expressed in kilodalton. Source data are provided as a Source Data file

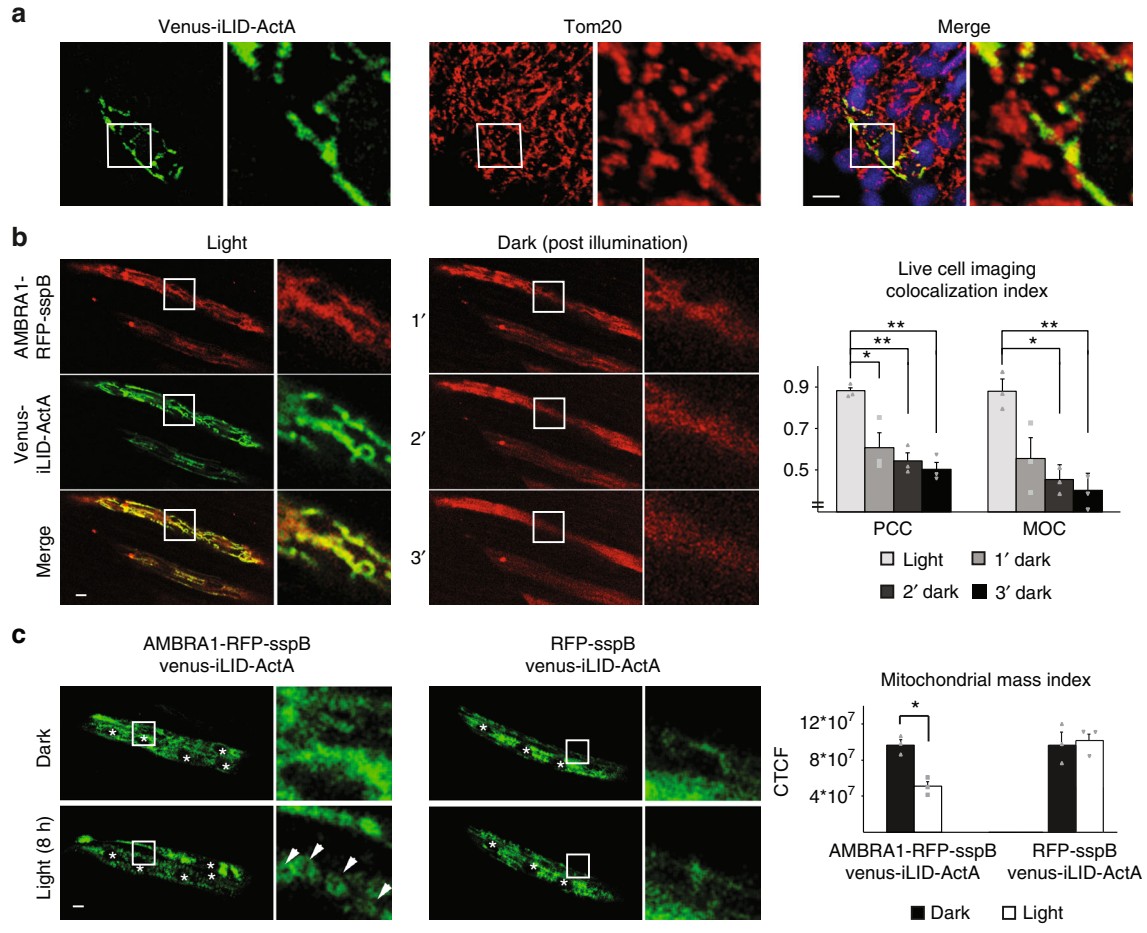

**Fig. 5** In vivo induction of mitophagy through an optogenetic bimodular system. **a** Zebrafish embryos were microinjected with a Venus-iLID-ActA overexpressing plasmid, then fixed 48-hpf and whole mount immuno-stained for Tom20 (red). Nuclei were counterstained with DAPI (blue). Inset: ×4 magnification. Scale bar: 10 μm. **b** Venus-iLID-ActA/AMBRA1-RFP-sspB microinjected zebrafish embryos were illuminated with blue light and then kept in the dark for 3 min, mimicking the experiment shown in Fig. 3a. Double-positive muscle fibers were photographed in order to analyze Venus-iLID-ActA/ AMBRA1-RFP-sspB dynamic interactions. PCC and MOC of the red over the green signal were quantified in three independent experiments. Inset: ×6 magnification. Scale bar: 10 μm. Data shown: mean ± S.E.M. Hypothesis test: ANOVA test. $^*p < 5 \times 10^{-2}$. $^{**}p < 10^{-2}$. **c** Venus-iLID-ActA/AMBRA1-RFP-sspB or Venus-iLID-ActA/RFP-sspB (negative control) double positive zebrafish embryos were illuminated for 8 h with a pulsed (2 s light/2 min dark) blue light; muscle fibers were analyzed. Upon light stimulation, round-shaped mitochondria (arrowheads) were clearly visible in AMBRA1-RFP-sspB but not RFP-sspB positive fibers. Images are the sum of three frames Z-stack. Stars indicate nuclei. Graphs show the overall reduction per single fiber of the Venus-iLID-ActA signal intensity corrected for the background in three independent experiments. Inset: ×8 magnification. Scale bar: 10 μm. Data shown: mean ± S.E.M. Hypothesis test: ANOVA test. $^*p < 5 \times 10^{-2}$. Source data are provided as a Source Data file

AMBRA1-RFP-sspB relocalized to MOM. Conversely, a slight increase of HSP60 signal in PQ-treated cells was detected in the control experiment (Fig. 6b, right panel), this being probably due to upregulation of the protein related to mitochondrial ROS accumulation[35].

Finally, we assessed cell death during PQ treatment by WB assay of the apoptotic marker cleaved PARP (cl-PARP, Fig. 6c). As expected, PQ caused apoptosis in the dark (lane 2, left panel), while the concomitant presence of blue light (lane 4, left panel) strongly attenuated cell death; no reduction of cl-PARP was seen in control conditions (Fig. 6c, lane 2/4, right panel). This trend was also confirmed when cleaved caspase3, a different protein of the same pathway, was analyzed (Supplementary Figure 17). Altogether, these data confirmed a cytoprotective function of AMBRA1-RFP-sspB-mediated mitophagy during PQ intoxication.

## Discussion
Here, we describe an optogenetically driven tool to specifically and reversibly induce mitophagy in the absence of any

mitochondrial poisons. This is particularly relevant in physio-pathological conditions, making its use an attractive (and mostly unexplored) strategy to ameliorate the phenotype of diseases in which dysfunctional mitochondria are involved. Loss of mitochondrial quality control has been postulated to be the keystone of many human diseases, such as tumorigenesis, muscle atrophy, diabetes, and Alzheimer's Disease, but the strongest correlations have been established with PD[36]. The electron transport chain, most of all Complex I and III, has been shown to be severely impaired in PD sporadic cases, likely causing a huge increase in intracellular ROS levels. Accordingly, signs of oxidative stress (lipid and protein peroxidation, GSH depletion, mtDNA damage, and peroxynitrites accumulation) are validated hallmarks of PD[36]. Although the reason why mainly Substantia Nigra pars compacta dopaminergic neurons are targeted is not known, it is possible that there is a relation with their energy needs, given that they express a self-generated pacemaker activity in basal conditions and are metabolically hyperactive[36]. Accordingly, understanding the role that mitophagy plays in vivo in its homeostatic control of mitochondrial-derived ROS is an urgent need. To date,

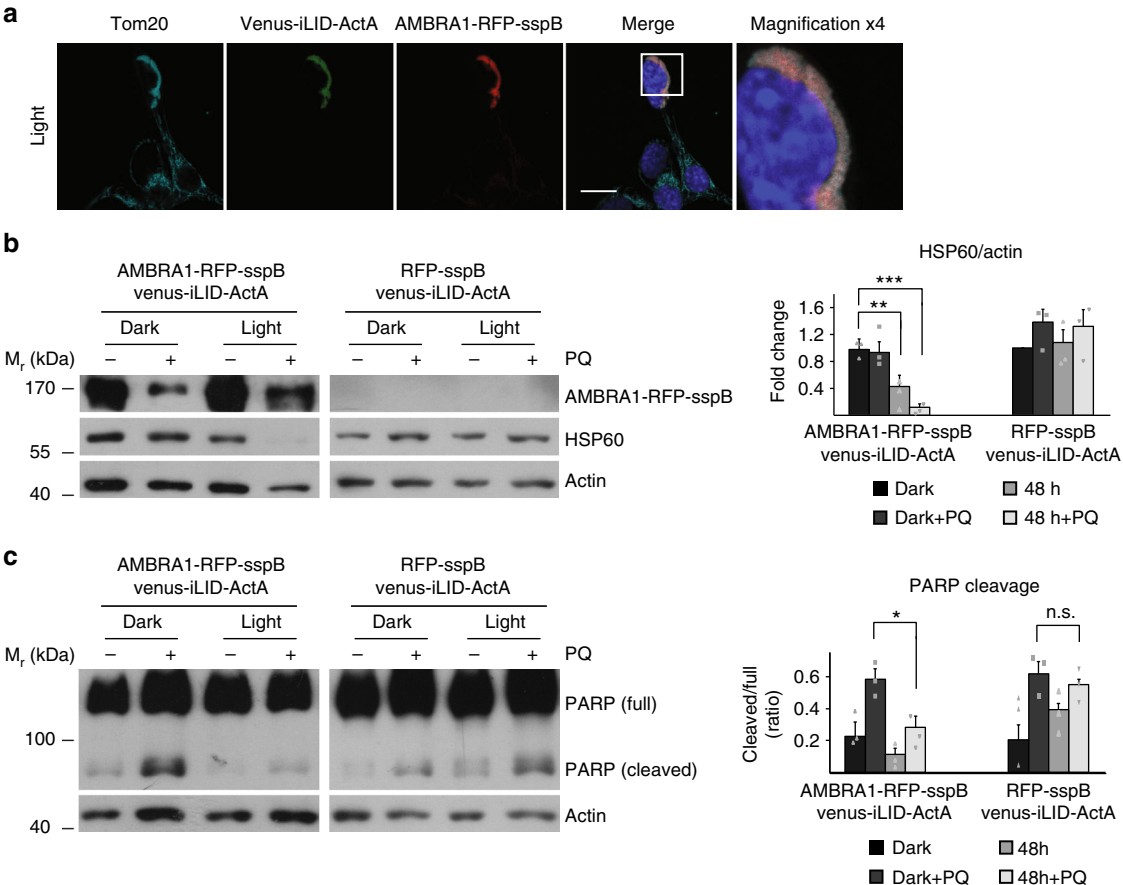

**Fig. 6** Optogenetic-mediated mitophagy prevents apoptosis in a model of neurotoxicity. **a** ETNA (Embryonic Telencephalic NAïve) cells co-expressing Venus-iLID-ActA/AMBRA1-RFP-sspB were irradiated with pulsed (1 s light + 1 min dark) light for 4 h, then fixed and stained for nuclei (DAPI, blue) AMBRA1 (red) and Tom20 (cyan). Inset: ×4 magnification. Scale bar: 10 μm. **b** ETNA cells, transfected with Venus-iLID-ActA/AMBRA1-RFP-sspB or Venus-iLID-ActA/RFP-sspB (negative control), were irradiated with pulsed blue light (1 s light + 1 min dark) for 48 h or left in the dark upon treatment with 250 μM PQ. Cell lysates were prepared and HSP60 levels (mitochondrial marker) explored in three independent experiments. Actin was used as a loading control. The graph reports the means of the normalized densitometric ratio between HSP60 and actin. Data shown: mean ± S.E.M. Hypothesis test: ANOVA test. $^{**}p < 10^{-2}$. $^{***}p < 10^{-3}$. $M_r$ (kDa): Relative molecular mass expressed in kilodalton. **c** Venus-iLID-ActA/AMBRA1-RFP-sspB or Venus-iLID-ActA/RFP-sspB (negative control) ETNA cells were handled as described in (**b**). WB was performed to reveal PARP protein in cell lysates. Upper band: full length PARP. Lower band: cleaved PARP. Actin: loading control. The graph summarizes the densitometric quantification of cleaved/full PARP in three independent experiments. Data shown: mean ± S.E.M. Hypothesis test: ANOVA test. $^*p < 5 \times 10^{-2}$. n.s. not statistically significant. $M_r$ (KDa): Relative molecular mass expressed in kilodalton. Source data are provided as a Source Data file

the high toxicity and the broad off-target effects of uncouplers and of other mitochondria poisons make it impossible to investigate this aspect. Remarkably, depolarizing agents' efficacy depends on the PINK1/Parkin axis, but these proteins are often mutated, both in sporadic and genetic variants of PD[37]. In line with that, as a proof of concept, we examined the cytoprotective role of AMBRA1-RFP-sspB-mediated mitophagy in a well-established experimental model of ROS-induced neurotoxicity. As expected, cell survival of PQ-treated proneural cells was strongly sustained (Fig. 6). In this sense, the possibility to stimulate this process in vivo (Fig. 5) opens new windows for research and therapeutic innovative strategies in neuroscience. In fact, AMBRA1-RFP-sspB/Venus-iLID-ActA bimodular system provides remarkable intrinsic advantages over any other mitophagy-based therapeutic candidates published so far.

First, AMBRA1-mediated mitophagy is Parkin dispensable[15]. This feature may hypothetically circumvent the problem that alterations of PINK1-Parkin pathway are often observed in neurodegenerative diseases[37].

Second, the quick temporal reversibility of the sspB/iLID binding permits to switch off the signal when undesired, and to finely tune the extent of the induction by regulating the time of illumination and/or frequency of spikes. Accordingly, we showed (Fig. 3) that AMBRA1 detachment from mitochondria restored normal levels of the mitochondrial mass in 24 h.

Third, optogenetics guarantees a high-spatial resolution, unattainable with other methods. It has been observed that just focusing the laser beam it is possible to push sspB/iLID binding in a subcellular fashion[19]. Thus, even if not demonstrated here, ideally the stimulation of mitophagy can be restricted to a single cellular compartment within polarized cells, such as neurons. We already reported that mitochondria from dendrites, but not cell somata, may contribute to the first stages of neurodegeneration in a mouse model of Alzheimer's disease[38]. Thus, it would be of great interest in the near future to stimulate mitophagy in subcellular zones and assess the resulting phenotype. It is also worth pointing out that in our conditions the iLID/sspB system can be used in living animals, and that we studied optogenetic dimerizers, as a whole, in functional assays in vivo.

To stress the versatility of the Venus-iLID-ActA/AMBRA1-RFP-sspB system we also demonstrated an effective mitophagy induction in human T cells collected from healthy donors. Since

mitophagy seems to be a key process in ROS control and cell survival of these cells[39], in the next future it would be highly intriguing to see, upon ex vivo reprogramming and reinfusion into patients, if an in loco boost of light-driven mitophagy could improve immune response in different hostile milieux, such as tumors or circumscribed sites of infection. It has also recently been demonstrated that T-cell mitochondrial dynamics have a huge impact on their metabolism and fate[40], this giving other perspectives to finely manipulate immunomodulation in specific sites and circumstances just by focusing a laser beam.

In sum, the AMBRA1-RFP-sspB/Venus-iLID-ActA bimodular system can represent a useful tool for several applications in various fields, either for basic research or as a therapeutic agent. Further understanding of the exact mechanisms by which AMBRA1 operates at mitochondria is still needed, in order to optimize the system and guide operators in future experiments.

## Methods

**Ethical aspects.** The authors declare that all relevant ethical regulations for animal testing and research have been complied with peripheral blood mononuclear cells (PBMC) were isolated from buffy coats obtained from healthy donors from Bambino Gesù Children's Hospital (OPBG) in Rome, Italy, who signed a written informed consent, in accordance with rules set by the Institutional Review Board of OPBG (Approval of Ethical Committee N° 969/2015 prot. N° 669LB). All husbandry and experimental procedures involving *D. rerio* embryos complied with European Legislation for the Protection of Animals used for Scientific Purposes (Directive 2010/63/EU).

**Plasmids and subcloning.** pLL 7.0 Venus-iLID-ActA and pLL 7.0 tgRFPt-sspB$_{micro}$ were purchased from Addgene (Cambridge, UK, #60413 and #60416, respectively). We took advantage of an AgeI unique restriction site upstream of tgRFPt-sspB$_{micro}$ to insert human AMBRA1 DNA sequence. Thus, AMBRA1 was PCR amplified through high fidelity PfuTurbo Polymerase (Stratagene, La Jolla, USA) from our pLPCX-AMBRA1 vector adding AgeI sites in the two primers (AMBRA1_F: ATTAACCGGTCGCCACCATGAAGGTTGTCCCAGAAAAG; AMBRA1_R: ATTAACCGGTCCACCGCCACCCCTGTTCCGTGGTTCTCC. The forward primer contained a strong Kozak sequence (GCCACC) upstream the first Methionine. Moreover, a small linker (Gly-Gly-Gly-Gly-Pro-Val-Ala-Thr) was inserted between the C-terminus of AMBRA1 and the N-terminus of tgRFPt to guarantee flexibility and prevention of sterical artefacts. Lastly, to ensure sustained expressions of transgenes, cDNAs were subcloned downstream a strong CMV promoter and upstream a WPRE post-transcriptional regulatory element. After subcloning, sequence and frame were confirmed by sequencing in the certified laboratory Eurofins Genomics (Ebersberg, Germany).

**Cell cultures and transfections.** HeLa (catalog number 93021013, Sigma-Aldrich, St. Louis, USA), HEK293T (ECACC 12022001) and ETNA cells (in-house cell line derived from murine E14 striatum primordia neurons and immortalized with a retrovirus transducing the tsA58/U19 large T antigen) were cultured in Dulbecco's Modified Eagle's Medium (DMEM, Gibco, USA) supplemented with 10% fetal bovine serum (FBS, ThermoScientific, Pittsburgh, USA) under 5% $CO_2$ at 37 and 33 °C, respectively. Cotransfections were performed using a 1:3 ratio of total plasmids DNA over Lipofectamine2000 (ThermoScientific) in Opti-MEM (ThermoScientific) for 6 h. HUWE1 interference was achieved through transient transfection of the SiRNA-HUWE1 5′-GCAGAUAAAUCUGAUCCUAAACCTG-3′ 3′′-UUCGUCUAUUUAGACUAGGGAUUUGGAC-5′ (Integrated DNA Technologies #150971213) by means of Lipofectamine 2000 according to the supplier's instructions. Prior all irradiation experiments, cell media were discarded and replaced by the phenol red-free FluoroBrite DMEM (ThermoScientific) added with 10% FBS, 2 mM L-glutamine (ThermoScientific) and 1% PenStrep (ThermoScientific). Cell treatments encompassed 40 mM NH₄Cl for 24 h, O/A [2.5 μM oligomycin (Calbiochem)/0.8 μM Antimycin A (Sigma-Aldrich)], supplemented with 20 μM QVD (Sigma-Aldrich) for 24 h or 250 μM Paraquat (Sigma-Aldrich) for 48 h. In order to block IKKα activity, cells were treated with the irreversible inhibitor BAY-117082 at 1 μM for 24 h, during illumination.

The 293T/17 packaging cell line (ATCC® CRL-11268) was cultured with IMDM (Gibco) supplemented with 10% FBS (Thermo Scientific) and 2 mM GlutaMax (Invitrogen, CA, USA). Cells were maintained in a humidified atmosphere containing 5% $CO_2$ at 37 °C. Cell lines were routinely tested for mycoplasma and authenticated by short tandem repeat analysis in the certified laboratory Eurofins Genomics.

**Illumination procedures.** Cells were illuminated inside a standard cell incubator at an irradiance of 500 μW per cm$^2$ with a 7.2 W low-energy blue light LED irradiator plugged to a high accuracy (error rate: 0.1 s per cycle) intervalometer (Model 451,

GraLab, Centerville, OH, USA). Cells were routinely irradiated with cycles of 1 s of light spaced by 1 min of dark resting state.

**Cell lysis and fractionation.** HeLa, HEK293T and ETNA total extracts were obtained as follows. After ×3 washing steps in ice cold phosphate-buffered saline (PBS) (Lonza, Basel, Switzerland), cells were lysed in homemade RIPA lysis buffer (150 mM NaCl, 50 mM Tris-HCl pH 7.5, 1% NP40, 0.5% sodium deoxycholate, 0.1% sodium dodecyl sulfate (SDS), 2.5 mM Na₃VO₄, 5 mM NaF, 1 mM DTT, all reagents purchased from Sigma-Aldrich) complemented with protease inhibitor cocktail (Sigma-Aldrich). Samples were collected and centrifuged at 13,000g for 10 min at 4 °C; supernatants were saved.

For cytosol/mitochondria fractionation, cells were lysed by mechanical breaking (~100 potter strokes) into a detergent-free buffer (0.25 M sucrose, 10 mM HEPES pH 7.5, 1 mM EDTA). A first centrifugation (600g for 10 min at 4 °C) was employed to pellet nuclei; subsequently, supernatants were recentrifuged at 12,000g for 15 min at 4 °C to separate a cytosolic upper phase from a mitochondrial pellet. The latter was eventually lysed in RIPA buffer.

All protein extracts were quantified through the Bradford Protein Assay (Bradford solution from Bio-Rad Laboratories) according to the manufacturer's protocols.

**Western blotting.** A 5 μg per sample were loaded on polyacrylamide gels and SDS-polyacrylamide performed. Gels were transferred onto PVDF membranes (Immobilion P, Millipore, Burlington, US) at 300 mA for 2 h. Membranes were subsequently blocked in 5% non-fat dry milk in TBS-T (50 mM Tris-HCl pH 7.5, 150 mM NaCl, 0.1% Tween-20) for 1 h at room temperature. Specific primary antibodies were used, followed by 3× 10 min washing steps in TBS-T and incubation with appropriate HRP (Horseradish Peroxidase)-conjugated secondary antibodies for 2 h at room temperature (1:3000 in blocking solution, Bio-Rad). Finally, bands were impressed on X-ray films (Aurogene, Rome, Italy) upon exposure of the chemiluminescent substrate ECL (Millipore) for 3 min. Primary antibodies used were: Rabbit anti-Actin (1:100,000, #4967 Cell Signaling Technology, Danvers, US), Mouse anti-AMBRA1 (1:3000, sc-398204 Santa Cruz Biotechnology, Dallas, USA), Rabbit anti-Casp3 (1:1000, #9662 Cell Signaling), Mouse anti-GAPDH (1:1000000, Mab374 Chemicon International, Temecula, USA), Rabbit anti-HSP60 (1:2000, sc-13966 Santa Cruz Biotechnology), Rabbit anti-PARP (1:1000, #9542 Cell Signaling), Rabbit anti-SOD2 (1:5000, ADI-SOD-110 Enzo LifeScience, Farmingdale, USA), Rabbit and Mouse anti-Tom20 (1:10000, sc-11415 and sc-17764 Santa Cruz Biotechnology), Mouse anti-Tubulin (1:50000, T6199 Sigma-Aldrich), Rabbit anti-HUWE1 (1:1000, A300-486A Bethyl Laboratories, Montgomery, USA). Uncropped and unprocessed blots are provided in the attached Source Data file

**Immunofluorescence.** Transfected or transduced cells grown on glass coverslips were washed and fixed with 4% formaldehyde (Sigma-Aldrich) in PBS for 15 min at 37 °C. After a permeabilization step (0.4% Triton X-100, Sigma-Aldrich, in PBS for 10 min at room temperature) cells were simultaneously incubated overnight at 4 °C with mouse anti-AMBRA1 (1:200, sc-398204 Santa Cruz Biotechnology) and rabbit anti-Tom20 (1:500, sc-11415 Santa Cruz Biotechnology) primary antibodies, dissolved in 3% Normal Goat Serum in PBS. Next day, cells were washed three times and treated with goat anti-mouse Alexa Fluor 555 (1:300, A-21425 Life Technologies) and donkey anti-rabbit Alexa Fluor 647 (1:300, A-31573 Life Technologies) secondary antibodies for 2 h at room temperature. Nuclei were counterstained with DAPI 1 μg per ml for 5 min, and then coverslips were mounted on polylysinated slides with an aqueous mounting medium (Sigma-Aldrich).

**Microscopy and image processing.** Immunofluorescence images represent a single-focus plane taken by a Zeiss LSM 700 confocal microscope (Carl Zeiss, Oberkochen, Germany). Live cell imaging was performed through a UltraVox Cellular Imaging Microscope (PerkinElmer, Waltham, USA) in a 37 °C thermostatic chamber at 5% of $CO_2$. Small adjustments in term of brightness and contrast were achieved through the software Adobe Photoshop CS6 (Adobe Systems, San Jose, US). The Area occupied by the signal was estimated through ImageJ. Corrected total cell fluorescence (CTCF) was calculated according to the following formula: $CTCF = ID − A \times B$, where ID stays for integrated density of single cells, $A$ for the area of the signal and $B$ for Integrated density of the background. The mean signal was calculated through ImageJ as the mean of ten different points per cell.

**Pulse-chase assay.** After 16 h of transfection, cells have been incubated for 4 h with AHA 50 μM (Invitrogen) in DMEM medium without Methionine (Gibco). Then cells have been washed twice with Dulbecco's PBS (Gibco) and illuminated (Light) or not (Dark) for 24 h. Cells have been fixed and permeabilized using Click-iT Cell Reaction Buffer Kit (Invitrogen) and conjugated with Alexa-647 alkyne (Invitrogen) following manufacturer instructions. AHA-alkyne-Alexa647 fluorescence have been measured with BD FACS-Celesta (using DIVA software) gated on Venus-positive cells.

**Retroviral constructs and production**. The expression cassette encoding for Venus-iLID-ActA, Ambra1-TagRFPt-sspB$_{micro}$ and the corresponding control TagRFPt-sspB$_{micro}$ were cloned inside a SFG retrovirus backbone. To produce the retroviral supernatant, 293T/17 were cotransfected with the retroviral vectors codifying for MoMLV gag–pol and the RD114 envelope, using the Genejuice transfection reagent (Merck Millipore, Germany), according to the manufacturer's instruction. Supernatant containing the retrovirus was collected 48 and 72 h later.

**Isolation and transduction of T lymphocytes**. PBMC were isolated from buffy coats obtained from healthy donors from Bambino Gesù Children's Hospital (OPBG) in Rome, Italy, who signed a written informed consent, in accordance with rules set by the Institutional Review Board of OPBG (Approval of Ethical Committee N°969/2015 prot. N° 669LB), using lymphocytes separation medium (Eurobio, Les Ulis, France). T lymphocytes were activated with immobilized anti-OKT3 (1 μg per ml, 14-0037-82 e-Bioscience, San Diego, USA) and anti-CD28 (1 μg per ml, 555726 BD Biosciences, Eysins, Switzerland) monoclonal antibodies in the presence of recombinant human interleukin-7 (IL7, 10 ng per ml; R&D, USA) and 15 (IL15, 5 ng per ml; R&D). Activated T cells were transduced on day 3 in 24-well plates pre-coated with recombinant human RetroNectin (Takara-Bio, Japan) using a specific retroviral supernatant. On day 5 after transduction, T cells were expanded in medium containing 45% RPMI1640 and 45% Click's medium (Sigma-Aldrich) supplemented with 10% FBS and 2 mM Glutamax and replenished twice a week.

**Cytofluorimetric analysis**. Expression of the CD3 cell surface molecule, the green and red fluorescence proteins were determined by flow-cytometry using standard methodology. Samples were analyzed with a BD LSRFortessa X-20. Data were analyzed using the FACSDiva software (BD Biosciences). For each sample, we analyzed a minimum of 20,000 events. For assessing apoptosis, AnnexinV Apoptosis Detection Kit has been used (eBioscience) on a BD Accuri C6 Cytometer (BD Bioscience). Gating strategies are shown in Supplementary Figure 18.

**Fish maintenance and embryos collection**. Zebrafish (D. rerio) of wild type strain were maintained at 28.5 °C in a Tecniplast acquarium system. Embryos were obtained by natural breeding and raised in Petri dishes containing Fish Water solution at 28.5 °C with a photoperiod of 14 h light/10 h dark. Developmental stages of embryos were determined according to the time after fertilization and morphological criteria.

**Microinjection of zebrafish embryos**. One- to two-cells stage embryos were microinjected into the yolk mass with 10 nl of a solution containing 30 ng per μl of pCS2 + hAMBRA1-RFP-sspB (or pCS2 + RFP-sspB as a control) and 15 ng per μl of pCS2 + Venus-iLID-ActA in ×1 Danieau's buffer. Injections were performed under a dissecting microscope using a microinjector attached to a micromanipulator (Leica Microsystems, Milan, Italy). Injected embryos were raised to the desired stages for in vivo imaging or collected for further analyses.

**In vivo imaging of zebrafish embryos**. Microinjected embryos were anesthetized with 0.04% tricaine (Sigma-Aldrich, E10521), embedded in 0.8% low-melting agarose prepared in fish water and lateral mounted on a depression slide. A Nikon C2 confocal system was used to record images that were analyzed with Fiji (ImageJ) software. Dynamic of interaction between AMBRA1 and mitochondrial outer membrane was studied by time-lapse confocal microscopy analysis of 32-hpf (hours post fertilization) zebrafish embryos treated with 488 nm wavelength laser (at 1/10 of the maximum intensity) for 2 s every 30 s over a period of 3 min. Acquisitions were made every 30 s in both green and red channels.

For functional experiments, microinjected embryos images were captured every 2 min over a period of 8 h from 34 hpf to 42 hpf with the same laser's settings.

**Immunofluorescence on zebrafish**. Totally, 48-hpf zebrafish embryos were fixed in 4% PFA overnight at 4 °C and then dechorionated. Embryos were stored in 100% MeOH at −20 °C, rehydrated in 1× PTw (1× PBS at pH 7.3, 0.1% Tween), incubated in 150 mM Tris-HCl at pH 9.0 for 5 min, followed by heating at 70 °C for 15 min. Subsequently, embryos were permeabilized with acetone for 20 min at −20 °C. After 3 h of incubation with blocking buffer (2% sheep serum, 1% bovine serum albumin, 1% DMSO in PBS), embryos were incubated for 3 days in primary antibody using 1:200 rabbit anti-Tom20 (sc-11415, Santa Cruz Biotechnology).

Alexa Fluor 555 goat anti-rabbit IgG (H + L) (A-27039, Invitrogen) was used 1:500 as secondary antibody for 2 h at room temperature. Nuclear counter staining was performed with the DNA binding dye Hoechst 33342. Images were taken using a Leica (Wetzlar, Germany) SP5 confocal microscope.

**Statistics**. All statistical calculations were performed and graphed using GraphPad Prism 6.01. Statistical analysis has been performed using two-tailed, paired (when the same cell was analyzed over time, as for the live cell imaging colocalization experiments) or unpaired (all other cases) Student's t test (two groups; 95% confidence interval) or with two-tailed one-way ANOVA with Tukey's post hoc test (multiple groups; 95% confidence interval). No other statistical tests were used.

Data are shown as mean ± SEM (single measurements indicated by dots and attached in the Source Data file). All experiments have been performed at least three times independently. No statistical methods were used to predetermine sample size. Pearson's and Manders' colocalization coefficients were automatically calculated using ImageJ software and the Jacop plug-in.

## Data availability

The authors declare that all data supporting the findings of this study are available within the paper and its supplementary information files. No datasets were generated in this study. The source data underlying Figs. 1b–d, 2b, c, 3a, b, 4a, b, 5b, c, 6b, c and Supplementary Figures 1, 3, 5, 7, 8b, 9, 10, 14, and 17 are provided as a Source Data file

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

## Acknowledgements

We thank Mrs. M. Acuña Villa and Dr. M. Bennett for secretarial and proofreading work respectively, Dr. C. Rodolfo, V. Turcanova, and M. Sorrenti for the technical support, Dr. L. Motta-Mena (Department of Biophysics, University of Texas) for kind suggestions. We also thank G. Milletti for the kind help in preparing Fig. 1a. Francesco Cecconi's laboratory is supported by grants from the Danish Cancer Society (KBVU R72-A4408, R146-A9364), the Novo Nordisk Foundation (7559, 22544), the Lundbeckfonden (R233-2016-3360), the LEO Foundation (LF17024). F. Cecconi's lab in Copenhagen is part of the newly established Center of Excellence for Autophagy, Recycling and Disease (CARD), funded by the Danmarks Grundforskningsfond (DNRF125). The Bjarne Saxhof Foundation also provided financial support. This research was also supported in part by grants from, FISM (2013 to F. Cecconi), the Italian Ministry of Health (Progetto Giovani Ricercatori GR2011-2012 to F. Strappazzon and Ricerca Corrente to I. Caruana), the Italian Ministry of Education, University and Research (Research Project of National Relevance 2017 ID 2017WC8499 to F. Locatelli), Roche (Roche per la ricerca 2017 to F. Strappazzon) and AIRC (Associazione Italiana Ricerca sul Cancro, 5x1000 ID 9962, AIRC IG 2018 ID 21724 to F. Locatelli, Start-up grant to I. Caruana and AIRC IG-2017 19826 to S. Campello).

## Author contributions

P.D'A. and F.C. had the ideas, designed the work plan, and wrote the paper. P.D'A. performed most of the experiments and all the cloning procedures. F.C. coordinated and supervised the work. P.D'A. and F.S. performed together most of the microscopy imaging. F.S. helped also in defining the strategies for the in vitro experiments. I.C., G.W., F. D.B., and F.L. provided human T lymphocytes, produced viral vectors, and infected the cells for the ex vivo experiments. G.M. and L.D.V. microinjected D. rerio embryos and performed the in vivo experiments. A.D.R. performed the experiments shown in Supplementary Figures 8, 9, 14, and in Supplementary Figure 10, the latter together with L.S. L.S. and S.C. performed the cytofluorimetric analysis in Supplementary Figures 5 and 10.

## Additional information

**Competing interests:** The authors declare no competing interests.

