## [Peer Review File · Nature Communications]

Reviewers' comments:

Reviewer #1 (Remarks to the Author):

The authors present very compelling data of a new system for inducing mitophagy in several cellular contexts. The data are very clear, and the technology appears very robust and tunable.

I think the manuscript would be greatly improved if either more experiments were done to show that known blocks in mitophagy impinge on their AMBRA-mediated events. This would help clarify, as the authors state at the end of the manuscript, "Further understanding of the exact mechanisms by which AMBRA1 operates at mitochondria is still needed, in order to optimize the system and guide operators in future experiments."

Minimally, much more detailed discussion is required to provide the clear impact of this new technique to understanding how mitophagy functions in human health and disease.

Reviewer #2 (Remarks to the Author):

D'Acunzo et al present in their manuscript a novel method to target by light induction AMBRA1 to mitochondria to induce mitophagy. They test their system in HeLa and murine neuronal cells and also in zebrafish. While the study is rather well controlled and most of the conclusions are convincing a comparison of this novel tool with known and established methods for mitophagy induction is missing. It would have been interesting to compare the mitophagy rate and dynamics of induction with other methods. I.e. are mitochondria faster and more efficiently removed using this method?

Major comments:

1. To my impression also under dark conditions significant amounts of AMBRA1 are already colocalizing with mitochondria. Can the induction of mitophagy under dark conditions be excluded?
2. The different amounts of plasmids in the transfected cells might be a major drawback compared to methods like the application of FCCP. The authors should show in how many cells mitophagy is really significantly induced.
3. In Fig.2B the SOD2 level is significantly increased (about 2.5 fold) in the dark upon treatment with NH₄Cl. This suggests significant lysosomal turnover of the marker protein even in the absence of induction by light. In general, the measurement of mitophagy is critical for the whole study. It would be better to use a pulse chase experiment instead of only looking at the steady state level of proteins.
4. The protein levels shown in Fig.2B and the reductions in Fig.2C does not really fit.
5. The reduction of Tom20 in microscopy after illumination is really impressive. Why does the Western blot only show 50% reduction?
6. Why is the level of cleaved PARP in the control panel of Fig.6C at the right side (dark conditions with paraquat) not as high as in the corresponding left panel?

Reviewer #3 (Remarks to the Author):

The manuscript by D'Acunzo and colleagues present a new protein construct that induces mitophagy upon blue-light illumination. They demonstrated reversible control of mitophagy and successfully applied it in cell lines, primary human cells, and in zebrafish. Overall, this is a very well executed and described study containing broadly applicable results. Publication in Nature

Communications is recommended, with the following minor modifications.

1. Throughout the study, the authors have used cells expressing RFP-sspB/Venus-iLID-ActA pair as a control. However, it is possible that the expression of RFP-sspB or Venus-iLID-ActA may generate toxicity. It would be useful to compare the SOD2 levels and mitochondria morphology of untransfected vs. expressing. They have compared cell viability of untransfected vs. RFP-sspB/Venus-iLID-ActA vs. AMBRA1-RFP-sspB/Venus-iLID-ActA cells in Supp. Fig. 5. A similar characterization for SOD2 levels would be useful.

2. Light sources and intensity need to be specified in the methods section. Also, it would be helpful to know the promoters used for expressing the constructs.

POINT-BY-POINT REBUTTAL LETTER

Reviewer #1

The authors present very compelling data of a new system for inducing mitophagy in several cellular contexts. The data are very clear, and the technology appears very robust and tunable.

I think the manuscript would be greatly improved if either more experiments were done to show that known blocks in mitophagy impinge on their AMBRA-mediated events. This would help clarify, as the authors state at the end of the manuscript, "Further understanding of the exact mechanisms by which AMBRA1 operates at mitochondria is still needed, in order to optimize the system and guide operators in future experiments."

We thank the Reviewer for helping us to better dissect this aspect, which, undoubtedly, was not defined in the previous version of the manuscript. Very recently, our lab has published a new paper¹ in which we thoroughly investigated the molecular mechanisms underlying AMBRA1-mediated and Parkin-independent mitophagy, clearly showing the crucial role of two novel players in AMBRA1-mediated events. More specifically, we demonstrated that the enzymatic activities of the E3 Ubiquitin Ligase HUWE1 and the kinase IKK α are strictly required for the AMBRA1-mediated mitophagy cascade; accordingly, we assumed that similar pathways are activated upon blue light illumination of AMBRA1-RFP-sspB/Venus-iLID-ActA co-expressing cells. In the new **Suppl. Fig. S14** of the revised manuscript, we now show that this is the case. Indeed, when AMBRA1-RFP-sspB/Venus-iLID-ActA illuminated cells are concomitantly silenced for HUWE1 or pharmacologically and irreversibly inhibited for IKK α kinase activity, no reduction in mitochondrial mass can be reported by Western Blot analysis of mitochondrial markers; indeed, this strongly suggests a halt in the specific process of AMBRA1-dependent mitophagy induction, as described in ¹

Minimally, much more detailed discussion is required to provide the clear impact of this new technique to understanding how mitophagy functions in human health and disease.

We are grateful to the Reviewer for this remark. We modified the discussion section adding a new detailed paragraph about the role of mitochondrial quality control (and

mitophagy, accordingly) in Parkinson's disease, which is the dysfunction the neurodegeneration more directly associated to mitophagy². We believe that the revised discussion is now improved.

Reviewer #2:

D'Acunzo et al present in their manuscript a novel method to target by light induction AMBRA1 to mitochondria to induce mitophagy. They test their system in HeLa and murine neuronal cells and also in zebrafish. While the study is rather well controlled and most of the conclusions are convincing a comparison of this novel tool with known and established methods for mitophagy induction is missing. It would have been interesting to compare the mitophagy rate and dynamics of induction with other methods. I.e. are mitochondria faster and more efficiently removed using this method?

We totally agree with the Reviewer's suggestion, although a direct comparison in the cell systems we have used in the paper is rather challenging. As extensively discussed in the Introduction paragraph, the most common method to induce mitophagy is the drastic treatment with uncouplers or a combination of Oligomycin/Antimycin (also known as OA treatment). However, as demonstrated by a plethora of papers (among others, both by us^{1,3} and by R. Youle group^{4,5}) wild type HeLa cells do not undergo mitophagy after treatment with uncouplers unless manipulated, e.g. by overexpression of Parkin or AMBRA1 itself. Moreover, given the high toxicity of these compounds, prolonged treatments (usually, more than 1h) shall be accompanied by pan-caspase inhibition to block cell death⁵, usually through the compound quinolyl-valyl-O-methylaspartyl-[2,6-difluorophenoxy]-methyl ketone (QVD). In the first place, we argued that these issues might potentially be confusing elements when comparing the effect of uncouplers to our more physiological system, and for this reason we decided to omit this analysis in the first version of the manuscript. However, in agreement with the Reviewer's concerns, we have now reconsidered our early assumptions. Indeed, the revised manuscript shows in the novel **Suppl. Fig. S9** a comparison of the mitophagy rate between 24h of illumination and 24h of OA+QVD treatment in HEK293T cells, which are fully Parkin-competent^{3,4}. No differences were found at this time point, suggesting that mitophagy maximal rate at the plateau is comparable between this two mitophagy-inducers. Admittedly, our intention is

not to propose a harder mitophagy induction method, but a more flexible, reversible and tunable assay (thus more suitable for pathway dissection studies)

Major comments:

1. To my impression also under dark conditions significant amounts of AMBRA1 are already colocalizing with mitochondria. Can the induction of mitophagy under dark conditions be excluded?

We thank the Reviewer for this concern, that helped us to better elaborate on that issue. Indeed, we cannot exclude that AMBRA1-RFP-sspB may also be present at the mitochondrial surface in a dark resting state. We have shown previously⁶ that endogenous AMBRA1 may partially be present at the mitochondrial surface in resting state where it interacts with mito-Bcl2, and that overexpression of myc-AMBRA1 increases the amount of the protein present at the MOM. However, in the absence of any mitochondrial insults, the overexpression of wild type AMBRA1 is not sufficient *per se* to induce mitophagy; by contrast, the expression of AMBRA1-ActA is able to trigger mitophagy in the absence of any other additional stimuli^{1,3}. This feature may be due to several reasons. It is possible, for example, that mitophagy is induced only when the amount of AMBRA1 at the MOM reaches a certain threshold. Alternatively, it is also possible that mitophagy induction strictly requires a chronic presence of AMBRA1 at the mitochondria for prolonged time (and thus, achievable only with ActA-anchored proteins in the absence of mitochondrial chronic dysfunctions), e.g. for recruiting HUWE1 to the MOM and to lead this E3 Ubiquitin Ligase towards its targets¹.

With this in mind, we believe that we can exclude any mitophagy induction in the dark resting state of AMBRA1-RFP-sspB overexpressing cells, mimicking what we have already published for the overexpression of myc-AMBRA1 in basal conditions. As extensively shown in **Fig 1c, 1d, 2c, 3a, 5b, 5c**, and in the new **Suppl. Fig. 8a**, we could not detect any mitochondrial morphologic abnormalities in the dark state. Moreover, the new **Suppl. Fig. 8b** shows more clearly that we could not report any reduction in mitochondrial markers when the dark state was compared to untransfected or to Venus-iLID-ActA/RFP-sspB cells. We hope that by this answer we have assessed the Reviewer's concern.

2. The different amounts of plasmids in the transfected cells might be a major drawback compared to methods like the application of FCCP. The authors should show in how many cells mitophagy is really significantly induced.

We fully agree with the Reviewer's concern. According to that, it has been reported that CCCP is able to trigger mito-aggregates formation after 1h of treatment⁴, while we have shown in **Suppl. Fig. 3** that after 1h of blue light stimulation ~50% of double-transfected cells appeared still normal (graph on the left), most likely because of protein dosage at the MOM, with the more transfected cells having a faster kinetics. However, after prolonged illumination (4h) the percentage dropped down to only ~30% (**Suppl. Fig. 3**), while virtually all cells analysed in this study showed some signs of mitochondrial abnormalities after 24h of pulsed blue light (indeed, this is confirmed by the absence of any difference after 24h between AMBRA1-RFP-sspB-mediated mitophagy and OA in **Suppl. Fig. S9**) These data suggest that in our system a prolonged time of illumination may overcome moderate expression of the transgenes at the single-cell level, given the massive re-localization to the MOM upon blue light stimulus. Given that, we cannot exclude a stronger effect of uncouplers at short time points (which is undoubtedly probable).

3. In Fig.2B the SOD2 level is significantly increased (about 2.5 fold) in the dark upon treatment with NH₄Cl. This suggests significant lysosomal turnover of the marker protein even in the absence of induction by light. In general, the measurement of mitophagy is critical for the whole study. It would be better to use a pulse chase experiment instead of only looking at the steady state level of proteins.

We agree with the Reviewer's concerns about the SOD2 levels upon treatment with NH₄Cl, which increased up to ~2.5 fold in all conditions. Given that SOD2 (SuperOxide Dismutase type 2) is a stress-sensitive protein, as a possible explanation, we have hypothesized an mRNA upregulation of the protein during the 24h of NH₄Cl treatment, taking into account the likely concomitant upregulation of intracellular stress pathways when autophagy is chronically blocked for 24h. Although in a first instance the two other markers analysed in this panel (Tom20, HSP60), which were not increasing upon NH₄Cl treatment, supported our conclusions, we are grateful to the Reviewer's elegant suggestion to fix the issue, which definitely has increased the strength of our work. In fact,

we have performed a pulse-chase experiment using the Methionine-analogue AHA (L-Azidohomoalanine) and found a significant decrease of the AHA levels when compared to the control after 24h of blue light illumination, this suggesting ongoing autophagy (new **Suppl. Fig. 10**). Remarkably, the long chase time in our conditions (24h) guaranteed that the analysis was exclusively addressed to the long-lived proteins proteolytic rate, which is considered a golden standard in measuring autophagic proteolysis⁷. Among the others, many mitochondrial matrix proteins, are, indeed, long-lived proteins, with a half-life that can last up to 24 days in rodents^{8,9}.

4. The protein levels shown in Fig.2B and the reductions in Fig.2C does not really fit.

Good point. In order to explain this apparent discrepancy, we hope the Referee will share our explanation as follows. The reduction shown in **Fig.2B** is a 2-fold order of magnitude (up to ~50%) while the one shown in **Fig.2C** is roughly a 3-fold order of magnitude (from $8.9 \cdot 10^5$ to $2.7 \cdot 10^5$, in average). However, while the **Fig. 2C** shows a graph referred to a 100% double-transfected population, a Western Blot lysate inevitably gives a more diluted result, given that a variable, heterogeneous population of double-, single- and un-transfected cells is considered. In our case, assuming a ~65% rate of double-transfected cells, which may still be considered a robust transfection efficiency (considering that AMBRA1-RFP-sspB is a ~180 KDa protein and a huge plasmid, accordingly), would be sufficient to explain this difference. In fact, $(8.9/2.7) \cdot 0.65 = 2.1$ fold, in line with the Western blot data

5. The reduction of Tom20 in microscopy after illumination is really impressive. Why does the Western blot only show 50% reduction?

In line with what discussed in Point 4, in the figure legends of **Suppl. Fig. 15**, we reported a double-infection efficiency of ~42%. On the other side, the absolute numbers for the immunofluorescence are ~32 A.U. and ~7.6 A.U. for the dark and light conditions, respectively, for a ratio of 4.2. If we normalize for the infection efficiency, the ratio becomes $4.2 \cdot 0.42 = \sim 1.8$ fold reduction, which is in line with the 50% reduction rate (2 fold) shown in the Western blot

6. Why is the level of cleaved PARP in the control panel of Fig.6C at the right side (dark conditions with paraquat) not as high as in the corresponding left panel?

Again, a nice catch. Indeed, Fig. 6C is formed by two hemi-panels (experiment on the left // control condition on the right) from two different gels, as indicated by the presence of a white space between them; in general, a direct comparison between them may be rather challenging (due to exposure time of the films, transferring issues etc.). However, we believe that in this specific case the difference may be due simply to technical reasons, e.g. a natural slight difference in transferring efficiencies between the two gels and the PVDF membranes. In fact, the fainter signal at the right side detectable not only in the case of dark conditions with Paraquat, but also in the other control lanes (the first lanes of both panels, dark lane without Paraquat) as well as for the loading controls. This strongly suggests that fewer proteins are present on the surface of the PVDF membrane on the right as a whole, explaining the apparent discrepancy. However, driven by Reviewer's concerns, in order to check if this was the case, we normalized the signal of the cleaved PARP over the respective actin (loading and transferring control) for this specific gel, and found that the in the lanes "Dark+Paraquat" was negligible (1.17 vs 0.95) (see here

below).

Reviewer #3:

The manuscript by D'Acunzo and colleagues present a new protein construct that induces mitophagy upon blue-light illumination. They demonstrated reversible control of mitophagy and successfully applied it in cell lines, primary human cells, and in zebrafish. Overall, this is a very well executed and described study containing broadly applicable results. Publication in Nature Communications is recommended, with the following minor modifications.

1. Throughout the study, the authors have used cells expressing RFP-sspB/Venus-iLID-ActA pair as a control. However, it is possible that the expression of RFP-sspB or Venus-iLID-ActA may generate toxicity. It would be useful to compare the SOD2 levels and mitochondria morphology of untransfected vs. expressing. They have compared cell viability of untransfected vs. RFP-sspB/Venus-iLID-ActA vs. AMBRA1-RFP-sspB/Venus-iLID-ActA cells in Suppl. Fig. 5. A similar characterization for SOD2 levels would be useful.

This an excellent indication. Following the Reviewer's suggestions, we generated a new **Suppl. Fig. 8** in which we compared, as requested, untransfected vs. RFP-sspB/Venus-iLID-ActA vs. AMBRA1-RFP-sspB/Venus-iLID-ActA co-expressing cells, in terms of mitochondrial morphology and SOD2 levels. As expected, the only condition with statistically significant changes was the AMBRA1-RFP-sspB/Venus-iLID-ActA + blue light condition.

2. Light sources and intensity need to be specified in the methods section. Also, it would be helpful to know the promoters used for expressing the constructs.

We are very grateful to the Reviewer to have highlighted the lacking of light specifications in the Materials and Methods section. We have modified this section in full agreement to the Reviewer's indications

References to this Rebuttal Letter:

1. Di Rita, A. et al. HUWE1 E3 ligase promotes PINK1/PARKIN-independent mitophagy by regulating AMBRA1 activation via IKK α . *Nat Commun* **9**, 3755 (2018)

2. Keane, P.C., Kurzawa, M., Blain, P.G., and Morris, C.M. Mitochondrial dysfunction in Parkinson's disease. *Parkinsons Dis* 2011, 716871 (2011)
3. Strappazon, F. et al. AMBRA1 is able to induce mitophagy via LC3 binding, regardless of PARKIN and p62/SQSTM1. *Cell Death Differ* **22**, 419-432 (2015)
4. Narendra, D. et al. Parkin is recruited selectively to impaired mitochondria and promotes their autophagy. *J Cell Biol* **183**: 795–803 (2008)
5. Lazarou, M. The ubiquitin kinase PINK1 recruits autophagy receptors to induce mitophagy. *Nature* **524**: 309–314 (2015)
6. Strappazon, F. et al. Mitochondrial BCL-2 inhibits AMBRA1-induced autophagy. *EMBO J* **30**, 1195-1208 (2011).
7. Zhang, J. et al. Development of a novel method for quantification of autophagic protein degradation by AHA labelling. *Autophagy* **10**, 901–912 (2014)
8. Menzies, R.A. & Gold, P.H. The turnover of mitochondria in a variety of tissues of young adult and aged rats. *J Biol Chem* **246**:2425-9 (1971)
9. Chan, X. Mitochondrial protein turnover: methods to measure turnover rates on a large scale. *J Mol Cell Cardiol* **78**: 54–61 (2015)

REVIEWERS' COMMENTS:

Reviewer #1 (Remarks to the Author):

The authors have responded to my previous comments.

Reviewer #2 (Remarks to the Author):

The authors have excellently addressed all points. The manuscript should be accepted for publication.